# Molecular Characterization of Plasma HDL, LDL, and VLDL Lipids Cargos from Atherosclerotic Patients with Advanced Carotid Lesions: A Preliminary Report

**DOI:** 10.3390/ijms232012449

**Published:** 2022-10-18

**Authors:** Gabriele Nieddu, Elena Michelucci, Marilena Formato, Cristina Ciampelli, Gabriele Obino, Giovanni Signore, Nicoletta Di Giorgi, Silvia Rocchiccioli, Antonio Junior Lepedda

**Affiliations:** 1Department of Biomedical Sciences, University of Sassari, 07100 Sassari, Italy; 2Institute of Clinical Physiology, National Research Council, 56124 Pisa, Italy; 3Department of Biology, Biochemistry Unit, University of Pisa, 56126 Pisa, Italy

**Keywords:** carotid atherosclerosis, plaque vulnerability, lipoproteins, targeted lipidomics, sphingomyelin, phosphatidylethanolamine

## Abstract

Carotid atherosclerosis represents a relevant healthcare problem, since unstable plaques are responsible for approximately 15% of neurologic events, namely transient ischemic attack and stroke. Although statins treatment has proven effective in reducing LDL-cholesterol and the onset of acute clinical events, a residual risk may persist suggesting the need for the detection of reliable molecular markers useful for the identification of patients at higher risk regardless of optimal medical therapy. In this regard, several lines of evidence show a relationship among specific biologically active plasma lipids, atherosclerosis, and acute clinical events. We performed a Selected Reaction Monitoring-based High Performance Liquid Chromatography-tandem Mass Spectrometry (SRM-based HPLC-MS/MS) analysis on plasma HDL, LDL, and VLDL fractions purified, by isopycnic salt gradient ultracentrifugation, from twenty-eight patients undergoing carotid endarterectomy, having either a “hard” or a “soft” plaque, with the aim of characterizing the specific lipidomic patterns associated with features of carotid plaque instability. One hundred and thirty lipid species encompassing different lipid (sub)classes were monitored. Supervised multivariate analysis showed that lipids belonging to phosphatidylethanolamine (PE), sphingomyelin (SM), and diacylglycerol (DG) classes mostly contribute to discrimination within each lipoprotein fraction according to the plaque typology. Differential analysis evidenced a significant dysregulation of LDL PE (38:6), SM (32:1), and SM (32:2) between the two groups of patients (adj. *p*-value threshold = 0.05 and log_2_FC ≥ |0.58|). Using this approach, some LDL-associated markers of plaque vulnerability have been identified, in line with the current knowledge of the key roles of these phospholipids in lipoprotein metabolism and cardiovascular disease. This proof-of-concept study reports promising results, showing that lipoprotein lipidomics may present a valuable approach for identifying new biomarkers of potential clinical relevance.

## 1. Introduction

Atherothrombosis resulting from carotid plaque rupture/erosion is the main contributor to major acute clinical events (https://vizhub.healthdata.org/gbd-compare/ (accessed on 5 October 2022)) including stroke, which represents the second largest cause of mortality and the third largest cause of disability globally, being responsible for 11.59% of total deaths and 5.65% of total disability-adjusted life years (DALYs), respectively.

Although high LDL-cholesterol levels and low HDL-cholesterol levels are well-established risk factors for cardiovascular disease (CVD), several lines of evidence indicate that a residual risk exists in patients who do not fully benefit from statin treatment, suggesting the need to identify novel markers of plaque development and evolution toward instability, useful for selecting the most appropriate patient-centered therapy [1,2]. In the past twenty years, large-scale MS/MS-based technologies have been applied to purified lipoprotein fractions for unravelling their specific protein cargos, in association with CVD and some CVD-associated pathological conditions including kidney disease and type 1 and 2 diabetes mellitus (an updated list of references with the main results can be found in Supplementary Material 1 of [3]). As far as our research group is concerned, we applied shotgun proteomics to the analysis of plasma LDL and HDL, identifying some novel lipoprotein-associated proteins and showing specific signatures for atherosclerotic patients with different types of carotid plaque [3,4], sorted by ultrasonography into hypoechoic (types 1 and 2) or “soft” and hyperechoic (types 3, 4, and 5) or “hard”, according to Gray–Weale classification [5]. Indeed, the improvement of imaging techniques has allowed routine characterization and detection of the features of carotid plaque vulnerability, also providing predictive information in both symptomatic and asymptomatic carotid artery stenosis [6,7,8,9]. 

Although these studies have proven to be very informative about the numerous functions of each lipoprotein class in relation to CVD, additional information for risk evaluation of acute clinical events onset is currently still missing. Indeed, the multiple biological functions of lipoproteins, particularly of HDL, result from both protein and lipid components, whose alterations are responsible for dysfunctional particles [10].

In the last years, plasma lipidomics have been gaining momentum, as several lines of evidence have shown a relationship among specific plasma lipid species, atherosclerosis [11,12,13,14,15,16,17,18], and the onset of adverse clinical events [19,20,21,22,23,24]. However, only few studies on the association between biologically active lipids, specifically associated with their lipoprotein carriers, and CVD have been reported so far, as also recently reviewed by Ding and Rexrode [25]. This is probably due, at least in part, to the difficulties in applying time-consuming lipoprotein purification procedures requiring specific expertise to large scale studies. Most of the published studies dealt with HDL showing alterations of the phospho- and sphingo-lipidomes in type 1 diabetes [26], type 2 diabetes [27,28], obesity and metabolic syndrome [29,30], dyslipidemia [31,32], and experimental atherosclerosis [33,34]. Some of them reported normalization of the HDL lipidome in metabolic syndrome following either Pitavastatin treatment [35] or weight loss and physical activity [36]. In addition, changes in the LDL lipidome and a consequent reduction of cardiovascular risk were obtained with statin treatment [37] or following phytosterol and omega-3 diet supplementation [38]. The HDL and LDL lipidomes were also analyzed in relation to coronary artery disease (CAD) and acute coronary syndrome (ACS) [39,40]. Furthermore, the HDL phosphosphingolipidome was analyzed in a rare population of subjects with premature CAD having high HDL-cholesterol levels, evidencing distinct signatures with respect to healthy subjects [41]. Very recently, Wang and coworkers developed high-resolution methods to study the relationship between proteome and lipidome of lipoproteins that applied to the analysis of HDL from high-fat, high-cholesterol diet-fed rabbits and ACS patients [42]. Using this approach, they showed that the combined features obtained allowed for better discriminating ACS from healthy individuals than direct proteome and/or lipidome quantification alone. 

This study aims to characterize the specific lipidomic profiles of plasma HDL, LDL, and VLDL fractions purified by isopycnic salt gradient ultracentrifugation from atherosclerotic patients undergoing carotid endarterectomy. By applying SRM-based HPLC-MS/MS to the analysis of 130 lipid species, some specific LDL-associated markers of potential clinical relevance were found dysregulated in association with echographic features of carotid plaque vulnerability.

## 2. Results

### 2.1. Lipoproteins Purification

Besides some alternative purification methods such as free solution isotachophoresis and immunoaffinity or size exclusion chromatography, ultracentrifugation in high-salt media represents the most widely used approach [3]. Indeed, the different lipoprotein classes have been historically classified according to their buoyant density. We performed isopycnic salt gradient ultracentrifugation followed by a further step of fraction flotation by high centrifugal fields to obtain highly purified HDL, LDL, and VLDL fractions, as assessed by SDS-PAGE analysis [3,4]. In particular, the electrophoretic profiles of each fraction did not show any gross contamination by plasma proteins, particularly albumin. As expected, apolipoprotein B100 was the most abundant protein in both VLDL and LDL fractions, whereas apolipoprotein AI represented over 80% of HDL apolipoproteins. No HDL contamination by Apo B100-containing lipoproteins was evidenced, as demonstrated by the absence of Apo B100 at the top of the HDL lane (Figure 1).

### 2.2. Targeted Lipidomics

Targeted lipidomics were performed using a SRM-based HPLC-MS/MS method on HDL, LDL and VLDL lipids fractions extracted with 2:1 chloroform-methanol (*v*/*v*) following the Folch procedure [43]. Parameters used for SRM analysis are reported in Appendix A. Using this approach, one hundred and thirty lipid species belonging to cholesteryl ester (CE), ceramide (Cer), phosphatidylcholine (PC), phosphatidylethanolamine (PE), lysophosphatidylcholine (LPC), lysophosphatidylethanolamine (LPE), sphingomyelin (SM), triacylglycerol (TG), and diacylglycerol (DG) (sub)classes were compared (Appendix A). Quality control and data visualization were performed to assess total lipid content in samples before and after normalization, distribution for each lipid (sub)class, and variation coefficients (CV%) for single lipid species, as reported in Appendix A. 

As expected, HDL, LDL, and VLDL fractions differed from each other in terms of distribution of lipid classes, with few exceptions (Figure 2). Indeed, both LDL and HDL showed higher CE levels with respect to VLDL, according to their well-known metabolic roles as CE carriers to the tissues (LDL) or from the tissues to the liver in the reverse cholesterol transport (HDL), whereas VLDL, which is known to be the main plasma carrier of TG from the liver to the tissues, displayed the highest TG contents. 

We performed a comparative analysis within each lipoprotein fraction sorted according to the plaque typology (“soft” or “hard”), which was determined by ultrasonography. Using this non-invasive routine method, some key characteristics of the lesion such as the fibrous cap thickness, the lipid-core size, the presence of calcifications and/or ulcerations, as well as the degree of stenosis were evaluated, providing useful information on plaque vulnerability. Overall, plaques are defined as “soft” if predominantly hypoechoic (characterized by a large lipid core), or “hard” in the presence of hyperechoic features (mostly fibrotic and/or calcific). Hereafter, for the sake of simplicity, each lipoprotein fraction will be referred to as “hard” or “soft” according to the plaque typology (e.g., HDL “hard”).

Following sorting for the plaque typology, no differences were evidenced between “hard” and “soft” fractions, except for DGs, which were higher in LDL “soft” (Appendix A).

Supervised multivariate analysis showed that lipids belonging to PE, SM, and DG classes mostly contributed to discrimination within each lipoprotein fraction according to the plaque typology (Appendix A).

Differential analysis of lipid profiles among “hard” and “soft” fractions evidenced a significant dysregulation of three lipid species, namely PE (38:6), SM (32:1), and SM (32:2), between LDL “soft” vs. LDL “hard” (adj. *p*-value threshold = 0.05 and log_2_FC ≥ |0.58|) (Figure 3 and Appendix A). Details on relative abundances of the three lipid species significantly dysregulated between LDL “soft” and LDL “hard” are reported in Figure 4.

## 3. Discussion

In the last years, the complexity of the wide array of bioactive lipids carried by lipoproteins is beginning to reveal itself thanks to the emerging MS/MS-based technologies applied to lipidomics [25,44].

Glycerophospholipids and sphingolipids are important regulators of lipoproteins metabolism including the activities of the involved enzymes. Some of them are precursors to several bioactive metabolites including lysophosphatidylcholines and ceramides, which are also involved in cell signaling [45,46,47]. Indeed, dysfunctional sphingolipid metabolism has been implicated in CVD, highlighting the need to delve deeper into lipid biochemistry for a better understanding of the molecular basis of these pathologies [48].

SMs are major structural phospholipids that determine surface pressure in lipoproteins, enhancing rigidity and influencing the activity of enzymes involved in lipid metabolism. In this respect, it is known that SMs strongly inhibit lipoprotein lipase (LPL)-mediated lipolysis [49] and play a key role in determining the lipoprotein CE content by acting as physiological inhibitors of cholesterol esterification by lecithin-cholesterol acyltransferase (LCAT) [50,51], mainly by hindering the binding of the enzyme to the lipoprotein surface [52]. Furthermore, SMs inhibit (whereas Cers activate) the hydrolytic activity of sPLA2, which liberates arachidonic acid from the lipoprotein surface [53].

Interestingly, it has been shown that an increased SMs to PCs ratio makes LDL more susceptible to secretory sphingomyelinase action, leading to the formation of aggregated LDL with high atherogenic potential [54], whereas enrichment of PCs and SMs in HDL strongly influences the rate of reverse cholesterol transport at multiple steps in the process [55,56]. SM levels in HDL have been reported to change in different pathological conditions including CAD [57], in which a reduction has been observed, and hypertension [58], in which an increase has been reported.

Ruut et al., have found that aggregation-prone LDL were enriched in sphingolipids, modifiable and predictive of future cardiovascular deaths [59]. Interestingly, LDL are the main carrier of SMs in circulation (50 mol% of total plasma SMs) [60]. Regarding PEs, Dang et al. have found an important positive association between atherosclerosis extension and progression in an extensive analysis of plasma lipids in ApoE^−/−^ mice [12].

Phospholipids have also proved valuable prognostic markers of cardiovascular disease and total mortality in the Ludwigshafen Risk and Cardiovascular Health study [61].

Recently, we identified a set of plasma lipids composed of seven SMs and three PEs that, even under optimal cholesterol-lowering treatments, allow for discrimination between high-risk CAD patients and controls, suggesting a role for these lipid classes in disease development [17]. 

In the present study, we applied targeted lipidomics to purified plasma lipoprotein fractions and we report a dysregulation of LDL-associated PE (38:6), SM (32:1), and SM (32:2) between two groups of patients with “hard” or “soft” carotid plaques, homogenous in terms of lipid profile, glycemia, blood pressure, and pharmacological treatments, suggesting their usefulness as potential biomarkers of plaque vulnerability. In any case, these results are in line with the above-mentioned studies showing a contributory role for these phospholipids in lipoprotein metabolism and CVD. Procedures for lipoprotein purification are relatively time-consuming, and, hence, not immediately amenable to application to large casuistries. However, we demonstrated in this proof-of-concept study that targeted lipidomics on purified lipoproteins may have a distinctive advantage over other bulk measurements in the identification of new biomarkers of clinical relevance, which would otherwise be masked by other plasma components. Thus, the evaluation of lipid composition at the lipoprotein level could represent, in perspective, an important tool in precision medicine and diagnostics. Further efforts must be made to validate the obtained results on a large cohort of patients as well as to assess their potential usefulness in risk evaluation of acute clinical events onset.

## 4. Materials and Methods

### 4.1. Sample Collection

Lipoprotein lipidomics was performed on twenty-eight patients undergoing carotid endarterectomy, enrolled in previously published studies [4,62]. Plaques were classified as soft (n. 16), having hypoechoic features, or hard (n. 12), having hyperechoic features, according to Gray–Weale classification [5]. To reduce confounders, stringent criteria of eligibility for enrolment were used, allowing for the selection of two groups of patients to be as homogenous as possible. Informed consent was obtained before enrolment. Institutional Review Board approval was obtained. The study was conducted in accordance with the ethical principles of the current Declaration of Helsinki. The main clinical parameters of the two groups of patients are summarized in Table 1. Fasting blood samples were harvested before surgery and immediately centrifuged at 2000× *g* for 10 min at 4 °C to collect plasma fractions, which were stored at −80 °C until analysis.

### 4.2. Lipoprotein Isolation and Purity Assessment

Lipoprotein fractions were purified by isopycnic salt gradient ultracentrifugation (Figure 5) [3,4]. Briefly, 0.9 mL of plasma sample, brought to d = 1.3 g/mL with solid NaBr (472.2 mg NaBr/mL plasma), were gently overlaid with 2.1 mL of a d = 1.006 g/mL solution (0.6% NaCl) (Figure 5a), and centrifuged at 541,000× *g* for 3 h at 4 °C in a TL-100 series ultracentrifuge equipped with a TLA-100 fixed-angle rotor (Beckman Coulter, Indianapolis, IN, USA) (Figure 5b). Afterwards, VLDL (d = 1.006–1.063 g/mL), LDL (d = 1.063–1.19 g/mL) and HDL (d = 1.19–1.21 g/mL) fractions were collected and further purified by a second centrifugation step, performed at 541,000× *g* for 2 h in saline solutions at density 1.006, 1.063, and 1.21 g/mL (Figure 5c), respectively, followed by desalting and concentration using Amicon Ultra-0.5 mL centrifugal filter units (10 KDa MWCO, Merck-Millipore, Darmstadt, Germany). The degree of purity was assessed by sodium dodecyl sulfate polyacrylamide gel electrophoresis (SDS-PAGE), as previously described [3]. 

### 4.3. HDL, LDL, and VLDL Fractions Processing for HPLC-MS/MS Analysis

HDL, LDL, and VLDL fractions, stored at −20 °C, were thawed at room temperature and immediately subjected to lipid extraction [43]. Each sample was brought to a final volume of 150 μL with 150 mM NaCl, transferred in a 1.5 mL microcentrifuge tube and 600 μL of 0.0625 μM DMS (N,N-dimethylsphingosine (d18:1)), selected as internal standard (ISTD), in MeOH/CHCl_3_ 1/2 (*v*/*v*) were added. The biphasic solution thus formed was sonicated for 10 min at room temperature, incubated at 25 °C for 30 min at 1000 rpm in a Thermomixer Compact (Eppendorf, Hamburg, Germany), and then centrifuged at 16,000× *g* for 10 min at 10 °C in a Microcentrifuge Heraeus Biofuge Fresco (Thermo Scientific, Waltham, MA, USA). From each sample, an aliquot of the lower phase was transferred into a glass HPLC vial, and the HPLC-MS/MS analysis was performed immediately.

### 4.4. HPLC-MS/MS Analysis

An SRM-based HPLC-MS/MS method was used to analyze the lipid extracts. The HPLC system was a Nexera X2 (Shimadzu, Kyoto, Japan) and the mass spectrometer was a QTrap 5500 (SCIEX, Concord, ON, Canada) equipped with a Turbo V ESI source. Vials were put in a refrigerated autosampler at 5 °C and each sample was injected in duplicate (injection volume 0.5 μL). To prevent carry-over, two needle wash solutions were used: MeOH/i-PrOH 50/50 (*v*/*v*) and MeOH/CHCl_3_ 1/2 (*v*/*v*). Gradient elution at a flow rate of 0.2 mL/min was performed on a Kinetex column packed with a C8 phase (100 × 2.1 mm, 1.7 μm, 100 Å by Phenomenex, Torrance, CA, USA). Mobile phase A was MeOH/H_2_O/i-PrOH 50/45/5 (*v*/*v*) and phase B was MeOH/i-PrOH 50/50 (*v*/*v*), both containing 5 mM ammonium formate. The gradient elution program was: 0 min, 65% B; 0.5 min, 65% B; 10 min, 100% B; 15 min, 100% B; 15.1 min, 65% B; 20 min, 65% B. The column temperature was 45 °C. The mass spectrometer was set in positive ion mode and the operating conditions were as follows: curtain gas 30 psi, ion spray voltage 5 kV, probe temperature 200 °C, ion source gas 1 and gas 2 30 psi, declustering potential 100 V, entrance potential 10 V, collision cell exit potential 19 V, collisional gas N_2_. A study by direct infusion was previously carried out [17] in order to optimize the collision energies for each lipid class and to choose the relative product ions to be used in the SRM analysis (Appendix A). A dwell time of 20 msec was used for each transition Q1/Q3. Analyst Software 1.6.3 (SCIEX, Concord, ON, Canada) was used to collect data while lipid peak areas were calculated with MultiQuant 2.1 software (SCIEX, Concord, ON, Canada).

### 4.5. Lipidomics Data Analysis

Lipidomic experimental data were analyzed using the lipidr package (version 2.8.0) [63] of R software (version 4.1.1). Firstly, lipidomics data were subjected to quality control, then normalized using the Probabilistic Quotient Normalization (PQN) method (lipid areas were compared across samples for each lipid species and then corrected for dilution errors by determining a dilution factor for each sample) [64] and log_2_ transformed. 

Lipid species distributions for each lipid (sub)class were compared between different lipoprotein fractions, irrespective of plaque typology, as well as between the two groups of patients, within the same lipoprotein fraction. Pairwise comparisons between fractions (LDL vs. HDL, VLDL vs. HDL, VLDL vs. LDL) or between plaque typology (“soft” vs. “hard”) were conducted using the estimated marginal means followed by Bonferroni correction.

Considering the whole lipidomics profile, orthogonal partial least-squares discriminant analysis (OPLS-DA), a multivariate supervised method, was performed for each lipoprotein fraction type using the plaque typology as a grouping variable with the aim of revealing patterns in data and discovering lipid levels related to the two different groups of subjects. Therefore, differential analysis was conducted comparing lipid levels in “soft” and “hard” through a moderated t statistic implemented within the limma R package. *p*-values were corrected using the Benjamini-Hochberg procedure in order to minimize any type I error and thus the occurrence of false positives. Lipids were considered significant and differentially altered with an adjusted *p*-value < 0.05 and a fold change FC ≤ 1/1.5 or FC ≥ 1.5.

## Figures and Tables

**Figure 1 ijms-23-12449-f001:**
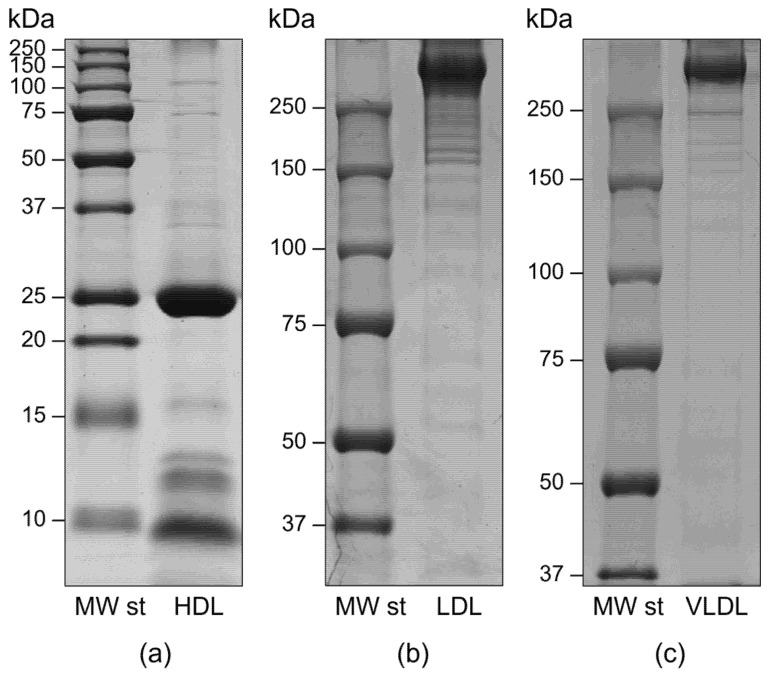
Representative mono-dimensional profiles of HDL (**a**), LDL (**b**), and VLDL (**c**) fractions purified by isopycnic salt gradient ultracentrifugation. Apolipoprotein profiles were obtained by SDS-PAGE in either 12% T (for HDL, under reducing conditions) or 6% T (for both LDL and VLDL, under non-reducing conditions) resolving gels.

**Figure 2 ijms-23-12449-f002:**
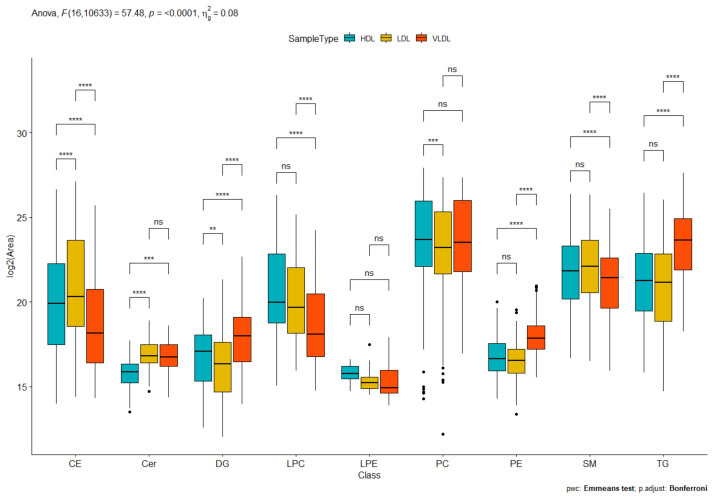
Box plot reporting the distribution of the nine different lipid classes (cholesteryl ester (CE), ceramide (Cer), phosphatidylcholine (PC), phosphatidylethanolamine (PE), lysophosphatidylcholine (LPC), lysophosphatidylethanolamine (LPE), sphingomyelin (SM), triacylglycerol (TG), and diacylglycerol (DG)) among HDL, LDL, and VLDL. ** *p*-value < 0.01, *** *p*-value < 0.001, **** *p*-value < 0.0001, ns not significant.

**Figure 3 ijms-23-12449-f003:**
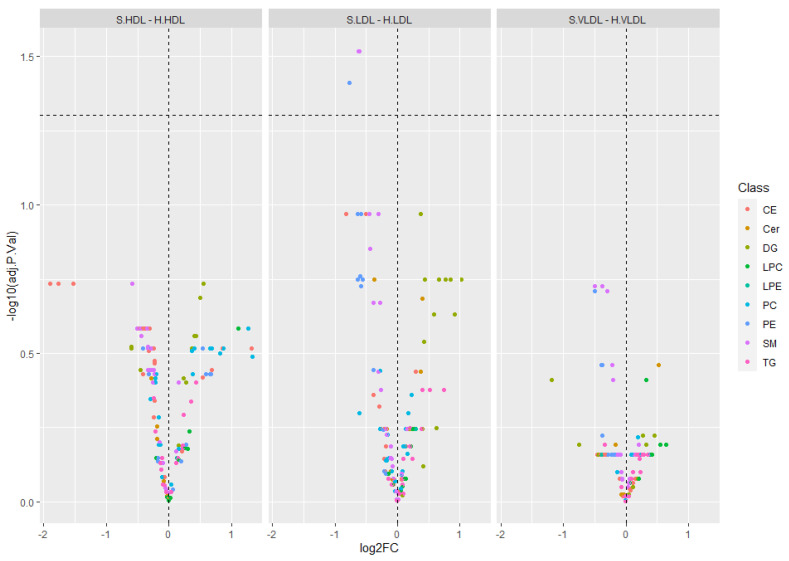
Volcano plots showing results of the differential lipid profile analysis (adj. *p*-value threshold = 0.05 and log_2_FC ≥ |0.58|) among “hard” and “soft” fractions. Lipids are colored according to the lipid class (CE, Cer, PC, PE, LPC, LPE, SM, TG, DG). PE (38:6), SM (32:1), and SM (32:2) were found significantly dysregulated in LDL “soft” vs. LDL “hard”. SM (32:1) and SM (32:2) are shown as a single purple spot (adjusted *p* value = 0.03 for both of them; Appendix A).

**Figure 4 ijms-23-12449-f004:**
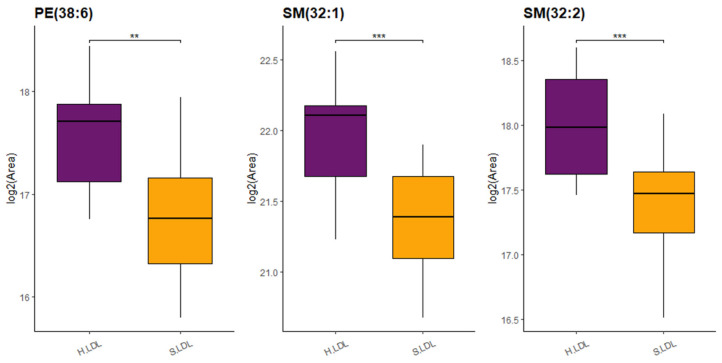
Boxplot reporting log_2_(Area) for lipids significantly dysregulated between LDL “soft” (S.LDL) and LDL “hard” (H.LDL): PE (38:6), SM (32:1), and SM (32:2) (** *p*-value < 0.01, *** *p*-value < 0.001).

**Figure 5 ijms-23-12449-f005:**
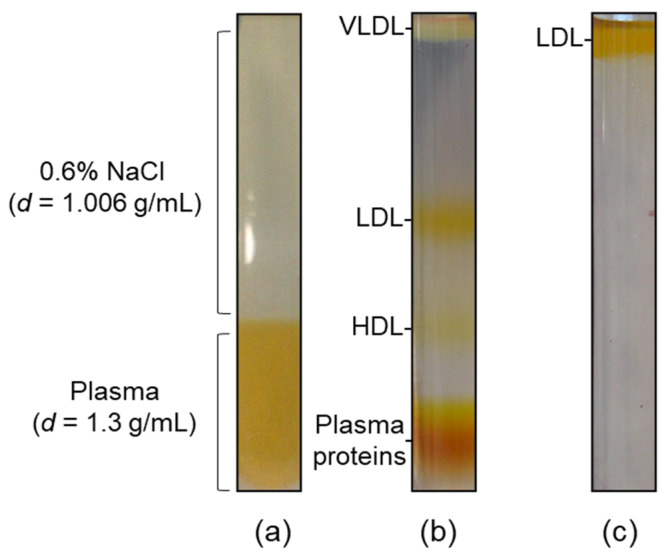
Ultracentrifugation procedure for lipoprotein purification. (**a**) Ultracentrifugation tube containing 0.9 mL of plasma sample added with NaBr, overlaid with 2.1 mL of a 0.6% NaCl solution; (**b**) self-generated density gradient following isopycnic ultracentrifugation, showing the three lipoprotein fractions under study; (**c**) LDL on the top of the tube following a washing step at d = 1.063 g/mL.

**Table 1 ijms-23-12449-t001:** The main clinical parameters of patients after sorting according to the plaque typology (hyperechoic or “hard” plaques and hypoechoic or “soft” plaques).

	“Hard” (*n* = 12)	“Soft” (*n* = 16)
**Age (Years) ***	66 ± 7.5	72.2 ± 7.3
**Male Gender**	8 (66.7%)	13 (81.2%)
**BMI**	26.5 ± 3.1	25.6 ± 3.4
**Triglycerides (mg/dL)**	102.1 ± 24	125.2 ± 43.8
**Total Cholesterol (mg/dL)**	190.3 ± 41.9	178.6 ± 47.4
**HDL Cholesterol (mg/dL)**	44.8 ± 9.3	44.1 ± 17.1
**nonHDL Cholesterol (mg/dL)**	145.5 ± 40.2	134.5 ± 46.1
**LDL Cholesterol (mg/dL)**	125.3 ± 37.5	106.8 ± 37.1
**TG/HDL-C ***	2.3 ± 0.4	3.2 ± 1.6
**Cholesterol Lowering Therapy**	9 (75%)	15 (93.8%)
**Glycemia (mg/dL)**	105.6 ± 14.3	115.6 ± 25.1
**HbA1C**	5.84 ± 0.472	6.482 ± 0.875
**Diabetes**	2 (16.7%)	5 (41.7%)
**Glucose Lowering Therapy**	1 (8.3%)	5 (31.2%)
**Systolic Blood Pressure (mmHg)**	139.1 ± 11.6	135.5 ± 19.4
**Diastolic Blood Pressure (mmHg)**	76.2 ± 11.2	72.2 ± 8.6
**Anti-hypertensive Therapy**	9 (75%)	12 (75%)

* *p* value < 0.05.

## Data Availability

Data produced in this study are available in the “Appendix A” or may be requested from the corresponding author.

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
