# Peer review of "Molecular Characterization of Plasma HDL, LDL, and VLDL Lipids Cargos from Atherosclerotic Patients with Advanced Carotid Lesions: A Preliminary Report"

_ijms, 2022, doi:10.3390/ijms232012449_

Round 1
Reviewer 1 Report
Nieddu and colleagues have reported on Molecular characterization of plasma HDL, LDL, and VLDL lipids cargos from atherosclerotic patients with advanced carotid lesions: a preliminary report. This is a potentially useful report for the readers.
Author Response
We thank the reviewer for his/her positive comments on this manuscript
Reviewer 2 Report
This is a proof-of-concept study whose originality is to focus on the different lipoprotein fractions in the plasma in relation to the phenotype of carotid atheroma plaques. The study is interesting and well conducted. The manuscript is pleasant to read, the objectives are well defined and the results clearly presented.
I have very few comments. Regarding the lipidomics analysis, it is stated that the lipid molecules are analyzed by subclasses (“Lipid species distributions for each lipid (sub)class”). Does this mean that the analyzed data represent the relative enrichment of the lipid molecules within their family and not the absolute amount of the molecules ?
Concerning the three identified molecules (PE(38:6), SM(32:1), and SM(32:2)), it could be interesting to present with more details the differences between the two groups using graphs. A more developped discussion on these three molecules could also be interesting. Are these molecules abundant within their family, what is their potential fatty acid composition. As example, PE(38:6) seems to be specifically altered as compared to other PEs, what are the most likely FAs in its structure (16:0 + 22:6 n-3) ?
Author Response
We thank the reviewer for his/her positive comments. Hereafter, a point-by-point reply.
I have very few comments. Regarding the lipidomics analysis, it is stated that the lipid molecules are analyzed by subclasses (“Lipid species distributions for each lipid (sub)class”). Does this mean that the analyzed data represent the relative enrichment of the lipid molecules within their family and not the absolute amount of the molecules ?
Although our analytical method would provide, if properly calibrated, absolute amount for each lipid, our focus was on the differences between “soft” vs “hard” within each fraction of HDL, LDL, and VLDL. Thus, no attempt to provide absolute values was performed.
The analysed data refer to the variation of each lipid between “soft” and “hard” groups (i.e. a given molecule was compared across the different samples). When referring to the entire (sub)class comparison, the total set of lipids belonging to that class was used to determine the (sub)class distribution in the three different lipoprotein fractions.
In all cases, areas were normalized before any processing by using Probabilistic Quotient Normalization (PQN) approach in order to correct for any dilution errors between samples (Figure S1A), as described in the method.
Concerning the three identified molecules (PE(38:6), SM(32:1), and SM(32:2)), it could be interesting to present with more details the differences between the two groups using graphs. A more developed discussion on these three molecules could also be interesting. Are these molecules abundant within their family, what is their potential fatty acid composition. As example, PE(38:6) seems to be specifically altered as compared to other PEs, what are the most likely FAs in its structure (16:0 + 22:6 n-3) ?
According to the reviewer suggestions, we added:
-a boxplot showing the three lipid species significantly dysregulated between LDL “soft” and LDL “hard” (figure 4 in the revised version);
-a table reporting the relative abundances of individual lipid species within each lipid class (supplementary table S2).
The parameters used in the SRM-MS analysis, and specifically the product ion type selected for each lipid subclass, are reported in the Table S1 of the Supplementary Materials. So far, the SRM method we developed does not allow a detailed characterization of the fatty acid composition for the monitored lipid species, neither in terms of acyl chain length, nor the double bond position along the side chains, nor the acyl chain position inside the molecular structure. For PE and SM subclasses we chose to monitor, respectively, the product ion types (precursor+ - phosphoethanolamine) and phosphocholine+. The rationale was to select the most intense peak to provide the best sensitivity even though some molecular information was lost. However, it is our intention to improve this SRM method by working on the choice of the product ion monitored, to gain more in-depth information on the fatty acid moiety.
Reviewer 3 Report
This study analyzed plasma lipidomic patterns using samples from twenty-eight patients undergoing carotid endarterectomy. The plaques were defined as either a “hard” or “soft” plaque by ultrasonography. The authors performed SRM-based HPLC-MS/MS analysis on plasma HDL, LDL, and VLDL fractions and found that PE(38:6), SM(32:1), and SM(32:2) in plasma LDL showed a difference between “hard” and “soft” plaques. Characterizing the plaque as unstable or stable is currently an important research topic. However, this preliminary analysis does not provide solid findings for the field.
1. Quantifying a plaque’s composition and characterizing the plaque as unstable or stable will generate more solid findings.
2. Features of carotid plaque instability are not enough to define a plaque more prone to rupture, such as missing criteria about necrotic core, fibrous cap, intraplaque hemorrhage, and neovascularization.
3. The lipidomic difference among HDL, LDL, and VLDL has been well studied. Lack of novelty.
Author Response
We thank the Reviewer for him/her comments. Hereafter, a point-by-point reply.
- Quantifying a plaque’s composition and characterizing the plaque as unstable or stable will generate more solid findings.
We partially agree with the reviewer. Indeed, histochemical and/or immunohistochemical plaque characterization provides detailed information on carotid intima-media thickness, ulceration and thrombosis, intraplaque haemorrhage, neovascularization, inflammatory cells and lipid content, as also shown in some of the previously published studies by our research group (Formato M, et al. Evidence for a proinflammatory and proteolytic environment in plaques from endarterectomy segments of human carotid arteries. Arterioscler Thromb Vasc Biol. 2004 Jan;24(1):129-35. doi: 10.1161/01.ATV.0000104013.71118.53. Epub 2003 Oct 30. PMID: 14592849; Lepedda AJ, et al. A proteomic approach to differentiate histologically classified stable and unstable plaques from human carotid arteries. Atherosclerosis. 2009 Mar;203(1):112-8. doi: 10.1016/j.atherosclerosis.2008.07.001. Epub 2008 Jul 12. PMID: 18715566; PMCID: PMC2659534; Lepedda AJ, et al. Protein sulfhydryl group oxidation and mixed-disulfide modifications in stable and unstable human carotid plaques. Oxid Med Cell Longev. 2013;2013:403973. doi: 10.1155/2013/403973. Epub 2013 Feb 4. PMID: 23431411; PMCID: PMC3575616). Nevertheless, ultrasonography, already adopted in some of our previous studies (Zinellu E, et al. Association between Human Plasma Chondroitin Sulfate Isomers and Carotid Atherosclerotic Plaques. Biochem Res Int. 2012;2012:281284. doi: 10.1155/2012/281284. Epub 2011 Dec 15. PMID: 22216412; PMCID: PMC3246695; Finamore F, et al. Apolipoprotein Signature of HDL and LDL from Atherosclerotic Patients in Relation with Carotid Plaque Typology: A Preliminary Report. Biomedicines. 2021 Sep 3;9(9):1156. doi: 10.3390/biomedicines9091156. PMID: 34572342; PMCID: PMC8465382) represents a widely used sensitive method for the detection and characterisation of vulnerable carotid plaques, providing also predictive information (see also reply to point 2). Therefore, we think that our findings are robust and deserve full consideration by the scientific community.
- Features of carotid plaque instability are not enough to define a plaque more prone to rupture, such as missing criteria about necrotic core, fibrous cap, intraplaque hemorrhage, and neovascularization.
We agree with the reviewer that features of plaque vulnerability assessed by ultrasonography are not diagnostic of a plaque prone to rupture but, there is plenty of evidence that carotid plaque echolucency provides predictive information of future stroke in both symptomatic and asymptomatic carotid artery stenosis. In this respect, the meta‐analysis by Gupta et al. (reference 7) on the association of ultrasonographic plaque echolucency with stroke risk in carotid atherosclerotic disease included a casuistry of 7557 subjects from 7 studies. As further support, we added the following meta‐analysis studies to the references:
-Brinjikji W, Rabinstein AA, Lanzino G, Murad MH, Williamson EE, DeMarco JK, Huston J 3rd. Ultrasound Characteristics of Symptomatic Carotid Plaques: A Systematic Review and Meta-Analysis. Cerebrovasc Dis. 2015;40(3-4):165-74. doi: 10.1159/000437339. Epub 2015 Aug 13. PMID: 26279159. [23 studies involving 6706 patients]
-Jashari F, Ibrahimi P, Bajraktari G, Grönlund C, Wester P, Henein MY. Carotid plaque echogenicity predicts cerebrovascular symptoms: a systematic review and meta-analysis. Eur J Neurol. 2016 Jul;23(7):1241-7. doi: 10.1111/ene.13017. Epub 2016 Apr 23. PMID: 27106563. [11 studies involving 8436 patients]
- The lipidomic difference among HDL, LDL, and VLDL has been well studied. Lack of novelty.
We completely disagree with the reviewer because this recent field of research is far from being exhaustively explored. Indeed, the current literature has shortcomings including the following:
-a very few studies addressed these issues. They have reported lipidome alterations in relation to some CVD-associated conditions including diabetes, obesity, metabolic syndrome, dyslipidemia, and acute coronary syndrome;
-so far, the studies have been carried out on a single lipoprotein fraction at a time, mainly on HDL. No more than a few have dealt with non-HDL particles;
-a very small number of participants were involved in these studies, thus hindering the transferability of results to the general population.
For the sake of completeness, the paragraph dealing with lipoprotein lipidomics in the “Introduction” has been enriched with some studies of interest to the aim of this manuscript [see references n. 28, 30-32, 34, 38-42 in the revised version]. As far as we know, this is the first study focusing on lipid composition of the three main fasting lipoprotein classes in association with carotid plaque vulnerability. We would be pleased to take into consideration any further published article suggested by the reviewer.
Round 2
Reviewer 3 Report
No more concerns.